# Coping, Social Support and Loneliness during the COVID-19 Pandemic and Their Effect on Depression and Anxiety: Patients’ Experiences in Community Mental Health Centers in Norway

**DOI:** 10.3390/healthcare10050875

**Published:** 2022-05-09

**Authors:** Zhanna Kan, Erik Ganesh Iyer Søegaard, Johan Siqveland, Ajmal Hussain, Ketil Hanssen-Bauer, Pia Jensen, Kristin Sverdvik Heiervang, Petter Andreas Ringen, Øivind Ekeberg, Erlend Hem, Trond Heir, Suraj Bahadur Thapa

**Affiliations:** 1Division of Mental Health and Addiction, Oslo University Hospital, 0424 Oslo, Norway; zhanka@ous-hf.no (Z.K.); ersoee@ous-hf.no (E.G.I.S.); p.a.ringen@medisin.uio.no (P.A.R.); oeekeber@online.no (Ø.E.); 2Division of Mental Health and Addiction, Institute of Clinical Medicine, University of Oslo, 0315 Oslo, Norway; trond.heir@medisin.uio.no; 3Division of Mental Health Services, Akershus University Hospital, 1478 Lørenskog, Norway; johan.siqveland@ahus.no (J.S.); ajmal.hussain@ahus.no (A.H.); ketil.hanssen-bauer@ahus.no (K.H.-B.); pijensen@icloud.com (P.J.); kristin.s.heiervang@ahus.no (K.S.H.); 4National Center for Suicide Research and Prevention, University of Oslo, 0315 Oslo, Norway; 5Division of Health Services Research and Psychiatry, Institute of Clinical Medicine, University of Oslo, 0315 Oslo, Norway; 6Centre for Medical Ethics, Institute of Health and Society, University of Oslo, 0315 Oslo, Norway; 7Department of Behavioral Medicine, Institute of Basic Medical Sciences, University of Oslo, 0317 Oslo, Norway; erlend.hem@medisin.uio.no; 8Institute for Studies of the Medical Profession, 0107 Oslo, Norway; 9Norwegian Centre for Violence and Traumatic Stress Studies, 0484 Oslo, Norway

**Keywords:** anxiety, depression, COVID-19, coping, Community Mental Health Centers, loneliness, social support

## Abstract

*Background:* Little is known about psychiatric patients’ experiences during the COVID-19 pandemic. The purpose of this study was to investigate associations of coping strategies, social support and loneliness with mental health symptoms among these patients. *Methods*: We recruited 164 patients from Community Mental Health Centers in June–July 2020. Participants responded to an online questionnaire on corona-related questions, Brief Coping Orientation to Problems Experience, Crisis Support Scale, a 3-item Loneliness Scale, and Hopkins Symptom Checklist-25. We used linear regression models to investigate associations between these and symptoms of depression and anxiety. *Results*: Almost 51% were aged 31–50 years and 77% were females. Forty-six (28%) participants reported worsened overall mental health due to the pandemic. The reported rates of clinical depression and anxiety were 84% and 76%, respectively. Maladaptive coping was independently associated with both depression and anxiety symptoms. Loneliness was independently associated with depression symptoms. *Conclusions*: Patients in Community Mental Health Centers in Norway reported high rates of depression and anxiety symptoms. Many of them reported worsening of their mental health due to the pandemic, even at a time when COVID-19 infections and restrictive measures were relatively low. Maladaptive coping strategies and loneliness may be possible explanations for more distress.

## 1. Introduction

The Corona Virus Disease 2019 (COVID-19) has caused a significant number of deaths and infected people all over the world since it was first discovered in China in December 2019 and later declared as a world-wide pandemic in February 2020 [1]. In Norway, the first case of COVID-19 infection was reported in February 2020. The Norwegian government, like many other governments, implemented several lockdown measures for few months from the middle of March to the middle of May. Unlike many countries, Norway never went into a complete lockdown. Even during this most restrictive period, all the groceries and public transports were open. People were asked to work from home in big cities like Oslo. Patients were given a choice of physical or online consultations. However, all the schools, kindergartens, Universities, restaurants, theaters were closed. Then, Norway gradually reopened with milder measures, until November 2020. The country has had one of the lowest infection rates in Europe and the lowest death rates due to the virus in the world [2].

Social distancing and lockdowns have been widely practiced to limit the spread of the virus [3]. Both imposed restrictions and fear of getting infected with COVID-19 have caused significantly reduced social interactions in everyday lives. Several studies including few prospective studies have reported high rates of stress and distress during this special time [4,5,6,7]. However, most of these studies have focused on mental health and distress among health care workers [6,8]. A cross-sectional study among frontline health care workers in Norway in March–April 2020 found the rates of depression, anxiety, and posttraumatic stress disorder (PTSD) to be 28.9%, 20.5% and 21.2%, respectively [9]. A population-wide poll from the USA in November 2020 found that self-reported mental health was the lowest they had ever measured [10].

Patients with pre-existing mental health conditions are particularly vulnerable and can react strongly to this pandemic since they are generally sensitive to distress and strain [11,12,13]. Psychological distress in patients can also be exacerbated by the loss of contact with health professionals and limited access to needed help and social contacts in the community. In the first period of COVID-19 pandemic in Norway, the capacity of mental health services was reduced for several reasons. Many consultations were through telephone or video. Social distancing and lockdowns are clearly risk factors for these patients [12]. Studies from earlier pandemics also show that individuals with psychiatric disorders and their family members suffer more in these situations [14,15,16,17]. An Indian study found that patients with severe psychiatric disorders, considered stable before the pandemic, had severe lack of information about COVID-19 and its mode of transmission. Thirty percent of these patients relapsed during the lockdown [18]. A similar finding was reported from a study in Spain [19].

Dealing with stress, including the stress caused by pandemics such as COVID-19, often depends on the use of specific coping strategies [20]. Coping is the individual’s cognitive, emotional, and behavioral ability to handle specific demands or burdens. Different coping strategies may constitute either a protective factor or a risk factor for anxiety and depression [21,22]. Coping strategies are often categorized as adaptive and maladaptive [22]. Adaptive coping strategies reduce the individual’s stress, while maladaptive coping fail [20]. There has been increasing interest to investigate coping and resilience to understand the impact of this pandemic in several settings [23,24,25]. For example, adaptive coping skills and social support have been found to protect against psychological distress in the general population in China [26]. Maladaptive coping strategies, such as using alcohol and drugs, were used to cope with pandemic distress in Canada [27].

Social support is another important factor that can enhance psychological adaptation to distress and psychosocial well-being. Social support is defined as information leading the individual to believe that he or she is cared for, loved, esteemed, valued, and belonging to a network of communication and mutual obligations [28]. Several studies have supported the importance of social support for mental health during this pandemic [20,22,29].

Other potentially negative consequences of the COVID-19 are the effects of social distancing and isolation on loneliness. Social isolation and loneliness have negative impact on cardiovascular and mental health outcomes [30]. Loneliness is caused not only by being alone, but also by being without some needed relationship or set of relationships. Loneliness and social isolation affect both psychological and physical health. Society may face heightened psychiatric morbidity and mortality risks, including suicide, as a result of burdensome loneliness [31]. A recent population-based study in Norway during the pandemic has reported a significant link between loneliness and depression and anxiety [32].

The purpose of the study was to explore levels of depression and anxiety during the COVID-19 pandemic in patients at Community Mental Health Centers (CMHC) in Norway; how they were related to coping strategies, social support, and loneliness. Specifically, the study investigated whether these patients were using adaptive or maladaptive coping strategies, and how coping was associated with anxiety and depression. Furthermore, we wanted to look at the independent effect of social support and loneliness in this situation.

Based on the above, we formulated the following hypotheses.

Psychiatric patients are reporting worsening of their mental health due to the pandemic.

Loneliness and using maladaptive coping strategies during pandemic are associated with more depression and anxiety symptoms.

Social support and adaptive coping strategies during pandemic are associated with lower depression and anxiety symptoms.

## 2. Materials and Methods

### 2.1. Study Design and Subjects

In this PsyCo-COVID-19 (Psychological distress and Coping in Patients in CMHC during COVID-19 pandemic) study, participants were recruited from patients treated in one of six participating CMHCs at Oslo or Akershus University Hospitals. These centers are serving a large portion of the population with mental health services in the greater Oslo Metropolitan area.

Patients registered in one of these CMHC’s outpatient or inpatient departments during the data collection period (June–July 2020) were informed about the study by their therapists. Those who accepted receiving more information were contacted through e-mail and text message with a link to information about the study and the written consent. The participants provided written consent through a secure and confidential digital personal identification system, which further led to the digital online questionnaire. Data was directly stored in the University of Oslo’s secured database for sensitive information called “Service for Sensitive Data” (TSD) [33].

Any adult patients over 18 years old in CMHCs were eligible for this study. The only additional inclusion criteria were proficiency in written Norwegian and ability to fill out the digital questionnaire. A reminder e-mail or text message was sent two to four weeks after the first invitation for those who had reported interest but had not completed the questionnaire.

### 2.2. Assessment Tools

#### 2.2.1. Background Questions

Our questionnaire included questions on socio-demographic variables, psychiatric and physical conditions, including infection with COVID-19, risk factors for severe COVID-19, being in quarantine due to contact with COVID-19 infected person or travel to a foreign country, or mandatory isolation due to COVID-19 infection.

#### 2.2.2. COVID-19 Specific Questions

The COVID-19 specific questions were for instance having experienced worsening of mental health due to the pandemic, need for more mental health care, worries about getting infected with COVID-19, worries for a family member getting infected, feeling isolated during the pandemic, and COVID-19 related changes in employment and economic situation.

#### 2.2.3. Coping

Brief-COPE (Coping Orientation to Problems Experienced) was used to measure coping during the COVID-19 pandemic. It consists of 28 items derived from the original COPE with 60 items and has demonstrated good psychometric properties in other COVID-19 related studies [21,34,35]. Each self-reported question is scored on a 4-point scale where 1 = “Not at all”, 2 = “a little bit”, 3 = “a medium amount”, and 4 = “a lot”. The 28 questions are normally grouped in pairs (score range 2–8) representing a total of 14 coping strategies, of which six are maladaptive and eight are adaptive [36]. The strategies normally considered maladaptive are self-distraction, venting, substance abuse, self-blame, denial, and behavioral disengagement. The adaptive strategies are acceptance, positive reframing, humor, emotional support, religion, active coping, instrumental support, and planning. Mean scores for maladaptive and adaptive strategies were used. Cronbach’s alpha was 0.79 for the maladaptive and 0.75 for the adaptive coping subscale.

#### 2.2.4. The Crisis Support Scale

The Crisis Support Scale (CSS) consists originally of seven items assessing social support. Each item is measured in a Likert-like scale from 1 (never) to 7 (always) [37]. To measure positive social support, the following five items were used: “Someone to listen to you”, “Contact with others in a situation”, “Ability to express oneself”, “Receiving sympathy”, and “Practical help”. Mean scores of the total were used. Cronbach’s alpha for this scale in our study was 0.85.

#### 2.2.5. Loneliness

The 3-item Loneliness Scale is an abbreviated version of the 20-item Revised UCLA Loneliness Scale [38,39]. The 3-item Loneliness Scale has been shown to have satisfactory reliability as well as concurrent and discriminant validity in previous research [38]. The scale assesses how often respondents feel that they lack companionship, feel left out, and feel isolated from others. Rated from 1 (hardly ever) to 3 (often), item sores are summed to create a sum score ranging from 3 to 9. Researchers in the past have used a cut-off score of 6 or higher as “lonely”, while a score of 3–5 normally is considered “not lonely” [40]. Cronbach’s alpha for the 3-item Loneliness Scale in our sample was 0.77.

#### 2.2.6. Depression and Anxiety

The 25-item version of Hopkins Symptoms Checklist (HSCL-25) was used to measure symptoms of depression (15 items) and anxiety (10 items). It is a self-reported measure, and it has been extensively used and found to be a psychometrically valid and a reliable indicator of anxiety and depression in both population-wide samples and in-patient populations in Norway and abroad [41,42]. Each item is rated on a four-point Likert-like scale (range 1–4) where 1 = “Not at all”, 2 = “A little”, 3 = “Quite a bit”, and 4 = “Extremely”. Mean scores of anxiety and depression subscales were calculated. We used the standard 1.75 as a cut-off to indicate possible clinical levels of anxiety and depression [41]. Cronbach’s alphas were 0.91 and 0.92 for the anxiety and depression subscales respectively in this sample.

### 2.3. Ethical Considerations

Participants were recruited through their therapists, who informed that study participation was voluntary and those who had consented and participated could withdraw from the study at any time. Written (digital) consent was mandatory before participation. Our coordinators at both the hospitals were also available if any of the participants had questions or wanted to talk with someone before, during or after their study participation. In addition, most of the participants were seeing their therapists regularly at the time of inclusion. The study was approved by the regional ethical committee of South-Eastern Norway (REK nr: 141152). Data protection officers at Oslo and Akershus University Hospitals also approved the study.

### 2.4. Statistical Analysis

Data was collected directly from the online/electronic forms filled in by the participants to the University of Oslo’s secure server (TSD). We used SPSS version 27 for our data analyses. Descriptive statistics with skewness and kurtosis values, Kolmogorov-Smirnov test of normality and histograms were used for assessing the normality of the distribution, linearity, and homoscedasticity for continues variables. The non-parametric Mann-Whitney U test was used when normal distribution of continuous scores was not assumed. Otherwise, independent samples *t*-tests were conducted to compare significant differences between groups. Correlation analyses using Spearman’s ρ was performed to explore relationship between both continuous and ordinal variables. Cohen’s standard was used to evaluate the correlation coefficient to determine the strength of the relationship. Cronbach’s alpha was computed to test internal consistency of the items used in psychometric scales.

Standard multiple linear regression analyses were performed to assess standardized coefficient β with 95% confidence intervals (CI) to determine associations with depression and anxiety symptoms. Variables selected as explanatory variables in the multiple linear regression analyses were based either on associations in bivariate regression analyses (*p* < 0.25) or literatures that have reported associations previously. The alpha level was set at *p* < 0.05.

## 3. Results

In summary, forty-six (28%) participants reported worsened overall mental health due to the pandemic. The reported rates of clinical depression and anxiety were 84% and 76%, respectively. Reported fear of getting infected or sick with the virus was low. Maladaptive coping was independently associated with both depression and anxiety symptoms. Loneliness was independently associated with depression symptoms.

### 3.1. Baseline Patient Characteristics

Of the 528 patients who received invitations and links to participate in the study, 164 responded (31%). The sociodemographic characteristics of the study participants are summarized in Table 1 below.

There were 77% female participants. Most patients were in the younger and middle-age groups; 83% were of ethnic Norwegian origin. The “not working” group (76%) included both unemployed and persons on different types of welfare, including paid sick leave. A significant portion of the participants (26%) reported worsening in their economic situation, but the participants did not report increased unemployment during the pandemic.

### 3.2. Self-Reported Somatic Health and COVID-19 Specific Questions

Only one participant had been infected with confirmed COVID-19 (0.6%). Forty (24%) participants reported that they had increased risk for severe COVID-19. Sixty-five (40%) reported being physically healthy.

Sixty-six (41%) reported feeling isolated during the pandemic. Forty-six (28%) reported worsening overall mental health due to the pandemic. Twenty-nine (18%) reported that they needed more mental health care because of the COVID-19 situation. Thirty (18%) participants had been in quarantine or mandatory isolation due to infection, contact with COVID-19 infected subjects, or travel to a foreign country. Sixty-one (37%) reported worries for a family member getting the virus, while only seven (4%) reported being afraid to get severely ill or die from the disease.

### 3.3. Coping Strategies

The most used maladaptive stress-coping strategy during the pandemic was self-blame (for instance criticizing oneself), used by 21%. Self-distraction by overly watching movies or TV, and overly turning to work or other activities were used by 18% and 11%, respectively. Refusing to believe that it has happened (0.6%), saying to oneself “this isn’t real” (2%), giving up the attempt to cope (3%), using alcohol or other drugs for feeling better (3%) or getting through the situation (3%) were less used maladaptive strategies.

Among the adaptive coping strategies, the most frequently used were accepting reality (38.4%) and learning to live with situation (18.3%), followed by thinking hard about what steps to take (16.5%), getting comfort, and understanding from someone (14%), getting emotional support from other people (10.4%) and concentrating efforts on doing something about the situation (10.4%). The least frequently used adaptive coping strategies were getting help/advice from other people (4.9%), positive reframing (5.5%), finding comfort in religion or spiritual beliefs (5.5%), praying, or meditating (6.7%), making fun of the situation (6.7%) and taking action to try to make the situation better (7.9%).

Maladaptive coping was significantly higher in participants who reported worsening of mental health due to the pandemic than those who did not (4.01 ± 0.88 vs. 3.20 ± 0.84, respectively; *p* < 0.001). There was no significant difference in adaptive coping between these groups (4.44 ± 0.71 vs. 4.25 ± 0.81, *p* = 0.168).

### 3.4. Positive Social Support

On the CSS, the average score was 20.7 ± 7.5 (range 5–35). Thirty-four percent reported that they never had contact with someone who had been in the same situation or had the same experience. CSS scores were significantly associated with loneliness (ρ = −0.39 (−0.54; −0.24); *p* = 0.001), non-Norwegian ethnical background (ρ = −0.20 (−0.36; −0.04); *p* = 0.014) and reported worsening of mental health due to the pandemic (ρ = −0.18 (−0.33; 0.01); *p* = 0.029).

### 3.5. Loneliness

The average total Loneliness score was 6.1 (±2.0). Seventy-three (44%) participants scored ≥ 6 on the loneliness measure and were in the “lonely” category while the remaining ninety-one (56%) were in the “not lonely” category. Approximately one third reported that they lacked companionship some of the time (*n* = 64 (39%)) or often (*n* = 53 (32%)). A similar proportion felt left out some of the time (*n* = 59 (36%)) or often (*n* = 48 (29%)) and felt isolated some of the time (*n* = 48 (29%)) or often (*n* = 67 (41%).

Examining correlations between some socio-demographic variables, CSS and COVID-19 related questions, loneliness showed medium association with reported worsening of mental health due to the pandemic (ρ = 0.32 (0.17; 0.46); *p* = 0.001) and CSS, as described above. This reflects more loneliness among those with an experience of greater mental impairments due to the pandemic and lower level of social support. Loneliness was also more common among younger participants (ρ = 0.16 (0.08; 0.01); *p* = 0.046). Not surprisingly, married participants (ρ = −0.26 (−0.41; −0.1); *p* = 0.002) and those living with somebody else (ρ = −0.27 (−0.41; −0.1); *p* = 0.001) reported less loneliness. Loneliness was not significantly associated with gender, education level, ethnical background, having children, employment, being quarantined or being in mandatory isolation.

### 3.6. Current Depression and Anxiety Symptoms

The mean depression score was 2.5 ± 0.7 and 137 (84%) had symptoms of depression at clinical level. Similarly, the mean anxiety score was 2.3 (±0.7), and 124 (76%) of the participants had symptoms of anxiety at a clinical level.

### 3.7. Associations between Coping Strategies, Social Support, Loneliness, and Current Psychiatric Symptoms

Two separate standard multiple linear regression analyses, for depression and anxiety symptoms respectively, were run based on findings from bivariate regression analyses or previously reported associations in earlier studies. Table 2 and Table 3 show factors associated with anxiety and depression, respectively.

Age above 50 was associated with lower anxiety scores. Reported worsening in mental health due to the pandemic and maladaptive coping during the pandemic was associated with higher anxiety scores. In the unadjusted model worries about getting infected, positive social support during the crisis and loneliness were also associated with anxiety scores, but the effects were no longer significant when adjusting for the other factors.

Age above 50 and social support were correlated with lower depression scores in the unadjusted model. Only the pandemic-related maladaptive coping strategies and more reported loneliness were associated with more depression symptoms when adjusting for other factors.

## 4. Discussion

### 4.1. How the Pandemic Affected Patients

This study examined how patients in CMHC reported current depression and anxiety symptoms and their coping strategies, social support, and loneliness during the COVID-19 pandemic in Norway. To the best of our knowledge, this is the first study that has looked at all these factors among patients in mental health services during the pandemic. Our results show that almost one third of the patients reported that their mental health was worsened by this pandemic and lock-down situation, even when infections and restrictive measures were relatively low. A study from China which compared psychiatric patients and healthy controls found similarly more worsening among the patients [11]. Another population-based study from USA and Canada also reported that those with prior mental health problems such as anxiety and depression reported more symptoms of COVID-19 related distress [43].

Patients in mental health services are likely to use maladaptive coping methods to mitigate the stress and challenges brought by the COVID-19 pandemic. We also found that maladaptive coping strategies under the pandemic were independently associated with more symptoms of depression and anxiety. In a study on adults in Australia during COVID-19, maladaptive coping such as self-blame, venting, behavioral disengagement and self-distraction were associated with poorer mental health [34]. Another study among students in Malaysia showed that students used maladaptive coping strategies more than adaptive coping strategies to deal with anxiety caused by the pandemic and the restriction of movement and socialization [22]. Another population-based study in Greece has similarly reported lower levels of depression among those using positive coping strategies during this pandemic [21].

Using adaptive coping strategies under the pandemic did not have significant associations with anxiety or depression symptoms, possibly because many of the adaptive strategies were rarely used. Similarly, positive social support did not play a role when adjusting for the other factors. Although adaptive coping and social support have been reported as protective factors for mental health [44], it should be taken into account that under pandemic with several restrictions it may be difficult to meet all the desired social supports as reported by few studies [45]. Furthermore, our sample consisted of selected patients referred to CMHC, perhaps because they were not able to use adaptive strategies on their own or get enough social support in their communities or primary care.

We did find that the feeling of loneliness during this pandemic was significantly associated with depression symptoms during the COVID-19 outbreak. An association between loneliness and depression has also been reported in one study among health care workers during COVID-19 [46] and another in a general population [47]. A recent population-based study in Norway also found a strong association between loneliness and depression during the pandemic [32]. Both the increased tendency for isolation and the reduction in CMHC services may have increased the loneliness in our population. This might be more damaging than the fear of getting infected or getting severe COVID-19, which was low in our sample.

### 4.2. Strengths and Limitations

We have used validated and commonly used instruments which were focused on symptoms and coping in a pandemic time point in our study. The exploratory study design makes it difficult to confirm the direction of associations or their causal relationship. The pre-pandemic data about mental health and coping behaviors in the study participants has not been thoroughly investigated. However, the explorative research is valuable as the understanding of the patients’ experiences is critical.

Self-report of worsening of mental health is the subject for bias and limitations. Nevertheless, we believe that the combination of the mentioned self-report with other validated and consistent instruments (HSCL-25, CSS, Revised UCLA Loneliness Scale and Brief-COPE) can still provide a more accurate picture of the subject.

There is still some likelihood of causal relationship between symptoms and pandemic in our study as the questions in the used instruments were focused on the pandemic related symptoms and coping strategies related to recent experiences—within last 2 weeks during the survey. Longitudinal studies may throw some light on the causal direction.

The use of electronic questionnaires might also have led to the relatively low response rate which may limit generalization of the findings. Some patients may have been sceptic or found it difficult to fill out online questionnaire, especially since we needed their personal identity number to get consent. Furthermore, there might have been relatively fewer men, and patients with immigrant background who responded than the average patient mix in these clinics. The participation rate of 77% of women in our sample is close to what we usually see in our clinical practice in these clinics as about 2/3 of our patients are females. Moreover, the study took place at the mid-summer vacation time when many patients were likely to have taken holidays that might also have affected the participation rate and participating population in this study. This underrepresentation of some groups of participants might have led to an underestimate of the mental health consequences during pandemic.

## 5. Conclusions

Many patients in Community Mental Health Centers in Norway reported that they got worse due to the pandemic, and that they have had high levels of both depression and anxiety symptoms. Maladaptive coping strategies and loneliness were associated with this. In a situation where infection pressure is low in the society, and social distancing measures are relatively mild, they may still pose a health burden in vulnerable groups such as patients in CMHC. In our sample, reported infection rates, fear of getting infected or fear of getting severe COVID-19 was low. Since many still reported worsening due to the pandemic and loneliness, other aspects of the situation, such as reduced mental health services and social distancing, may affect these groups.

### Implications

Maladaptive coping during the COVID-19 pandemic situation is related to self-reported increased depression and anxiety symptoms in our study. Thus, interventions targeted at identifying and treating maladaptive coping strategies could be a valuable part of treatment during a pandemic, regardless of primary diagnosis. Adaptive coping strategies, like seeking help and support from others was seldom used in this population and may be encouraged.

Other risk factors such as loneliness and experience of worsening of mental health due to the pandemic may be addressed by establishing alternative forms of social contacts and psychosocial help. In future research, a longitudinal intervention study on patient populations could see how this could affect the symptomatology and outcomes. Health authorities should consider allocation of more resources to psychiatric patients in general during and after the pandemic.

## Figures and Tables

**Table 1 healthcare-10-00875-t001:** Socio-demographic characteristics of 164 patients attending Community Mental Health Centers in the Oslo Metropolitan area during the COVID-19 pandemic.

	*n* = 164	%
*Age in years*		
18–30	65	39.6
31–50	83	50.6
51–70	16	9.8
*Gender*		
Men	38	23.2
Women	126	76.8
*Education, years of schooling*		
1–10	13	7.9
11–13	79	48.2
>13	70	42.7
Missing	2	1.2
*Marital status*		
Single	74	45.1
Married or living together	77	47.0
Divorced/widow or widower	13	7.9
*Number of children*		
None	109	66.5
1	27	16.5
2 or more	26	15.6
Missing	2	1.2
*Ethnical background*		
Ethnic Norwegian	136	82.9
Non-Norwegian	28	17.1
*Housing arrangement*		
Living alone	41	25.0
Living with somebody	123	75.0
*Current employment*		
Not working	124	75.6
Working	40	24.4

**Table 2 healthcare-10-00875-t002:** Factors associated with anxiety symptoms according to HSCL-25 (*n* = 161).

	Unadjusted			Adjusted		
Variables	β	95 CI	*p*	β	95 CI	*p*
Age > 50 years	−0.21	−0.87; −0.14	0.007	−0.18	−0.75; −0.07	0.018
Gender (women)	0.09	−0.12; 0.41	0.289	−0.02	−0.28; 0.21	0.786
Marital status (married)	−0.07	−0.32; 0.13	0.391	0.14	−0.006; 0.41	0.057
Worsening in mental health due to pandemic	0.44	0.46; 0.91	<0.001	0.23	0.11; 0.58	0.004
Worries about getting COVID-19 infected	0.18	0.03; 0.48	0.027	0.07	−0.09; 0.30	0.315
Maladaptive coping	0.55	0.33; 0.53	<0.001	0.33	0.13; 0.38	<0.001
Adaptive coping	0.16	−0.004; 0.29	0.057	0.07	−0.08; 0.19	0.399
Crisis support	−0.22	−0.04; −0.01	0.007	−0.11	−0.03; 0.004	0.169
Loneliness	0.41	0.09; 0.20	<0.001	0.16	−0.003; 0.11	0.063

Note R^2^ = 0.414.

**Table 3 healthcare-10-00875-t003:** Factors associated with depression symptoms according to HSCL-25 (*n* = 161).

	Unadjusted			Adjusted		
Variables	β	95 CI	*p*	β	95 CI	*p*
Age > 50	−0.16	−0.77; −0.02	0.038	−0.10	−0.53; 0.06	0.118
Gender (women)	0.11	−0.07; 0.46	0.150	−0.05	−0.30; 0.14	0.480
Civil status (married)	−0.11	−0.39; 0.07	0.163	0.07	−0.08; 0.28	0.264
Worsening in mental health due to pandemic	0.42	0.44; 0.89	<0.001	0.11	−0.03; 0.38	0.098
Worries about getting COVID-19 infected	0.102	−0.08; 0.38	0.200	−0.044	−0.24; 0.12	0.494
Maladaptive coping	0.58	0.35; 0.55	<0.001	0.41	0.20; 0.42	<0.001
Adaptive coping	0.10	−0.06; 0.24	0.222	0.002	−0.12; 0.13	0.975
Crisis support	−0.24	−0.04; −0.01	0.003	0.02	−0.01; 0.02	0.825
Loneliness	0.60	0.17; 0.26	<0.001	0.45	0.10; 0.21	<0.001

Note R^2^ = 0.524.

## Data Availability

The data presented in this study can be available on request from the corresponding author. The data are not publically available due to privacy and ethical issues.

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
