# Peer review of "Coping, Social Support and Loneliness during the COVID-19 Pandemic and Their Effect on Depression and Anxiety: Patients’ Experiences in Community Mental Health Centers in Norway"

_healthcare, 2022, doi:10.3390/healthcare10050875_

Round 1

Reviewer 1 Report

The authors investigated associations between coping strategies, social support, loneliness, and worsening of mental health during the COVID-19 pandemic among patients in community mental health services.

They recruited  hundred and sixty-four patients were recruited from Community Mental Health Centers in June-July 2020.

Their  questionnaires focused on corona-related questions, Brief Coping Orientation to Problems Experience, Crisis Support Scale, a 3-item Loneliness Scale, and Hopkins Symptom Checklist-25.

They highlighted that forty-six (28%) participants reported worsened overall mental health due to the pandemic. 84% reported rates over cut-off of possible clinical depression and 76% of clinical anxiety.

The outcome from the study associated maladaptive coping independently with both depression and anxiety symptoms. Loneliness was independently associated with depression symptoms.

The authors   concluded that: (a) Patients in Community Mental Health Centers in Norway reported high rates of depression and anxiety symptoms. (b) Many of them reported worsening of their mental health due to the pandemic, even at a time when COVID-19 infections and restrictive measures were relatively low. (c) Maladaptive coping strategies and loneliness may be possible explanations for more distress.

The manuscript is very interesting and written with enthusiasm. There is a lot of information useful and encouraging feature studies.

With pure academic spirit I propose the following improvements:

  1. Please dedicate some efforts to improve the abstract. It should better summarize the sections. It directly started with “We wanted to investigate….”. Introduce it with a sentence of background and better balance it. See also the following comment
  2. You say “We wanted to explore levels …….Specifically, we wanted to investigate whether these patients were using adaptive or maladaptive coping strategies, and how coping was associated with anxiety and depression. ..I’d like a more explicit purpose “the purpose of the study was to….Specifically the study investigated…”
  3. Table 1 should be moved in the first section of the methods
  4. Introduce the themes of the results by some a few sentences.
  5. Check the reference citation style. It should be [] and not ()
  6. Check the style used in the references. For example et al is not used in MDPI..,a web link need accedd on date…ecc.

7.. Tables do not follow the MDPI standard

Author Response

REVIEWER 1

  1. You say “We wanted to explore levels …….Specifically, we wanted to investigate whether these patients were using adaptive or maladaptive coping strategies, and how coping was associated with anxiety and depression. ..I’d like a more explicit purpose “the purpose of the study was to….Specifically the study investigated…”

-Thank you for your suggestion. We have also changed that on page 4 as follows.

“The purpose of the study was to explore levels of depression and anxiety during the COVID-19 pandemic in patients at Community Mental Health Centers (CMHC) in Norway; how they were related to coping strategies, social support, and loneliness. Specifically, the study investigated”

  1. Table 1 should be moved in the first section of the methods

-We think that it’s more usual to have this table in the result section. We have moved the table there.

  1. Introduce the themes of the results by some a few sentences.

-We have added the followings as the first paragraph under the Results.

“In summary, forty-six (28%) participants reported worsened overall mental health due to the pandemic. The reported rates of clinical depression and anxiety were 84% and 76%, respectively. Reported fear of getting infected or sick with the virus was low. Maladaptive coping was independently associated with both depression and anxiety symptoms. Loneliness was independently associated with depression symptoms.»

  1. Check the reference citation style. It should be [] and not ()

-We have changed brackets from ( ) to [ ].

  1. Check the style used in the references. For example et al is not used in MDPI..,a web link need accedd on date…ecc.

-We have now corrected for the references.

7.. Tables do not follow the MDPI standard.

-We have moved all the tables into the text.

Reviewer 2 Report

This is a well conducted study, solid literature review and sound methodology. However given the nature of the sudy and the selected country and demographic of the survey audience, a more detailed commentary of the measures taken by the government that impcted society. For example, in Austalia a 6 month lockdown during which leaving the home, even to visit family including a curfew impacted behaviour of citizens. What were the extent of the restrictive measures in Norway? 

The data sampling is relevant but the regulatory reponses are necessary to permit measurement against the impact assessed in other countries and regions. 

Author Response

This is a well conducted study, solid literature review and sound methodology. However given the nature of the study and the selected country and demographic of the survey audience, a more detailed commentary of the measures taken by the government that impcted society. For example, in Austalia a 6 month lockdown during which leaving the home, even to visit family including a curfew impacted behaviour of citizens. What were the extent of the restrictive measures in Norway? 

The data sampling is relevant but the regulatory reponses are necessary to permit measurement against the impact assessed in other countries and regions. 

-Thanks for your valuable comments. We have added the following in the introduction.

Unlike many countries, Norway never went into a complete lockdown. Even during this most restrictive period, all the groceries and public transports were open. People were asked to work from home in big cities like Oslo. Patients were given a choice of physical or online consultations. However, all the schools, kindergartens, Universities, restaurants, theaters were closed.”

Reviewer 3 Report

Many references are too old (14,15,16,17,30,36,37,38,39,40,41,42)

if possible find more recent

Author Response

Many references are too old (14,15,16,17,30,36,37,38,39,40,41,42)

if possible find more recent

-Thanks for your suggestions. Except for reference 30 which we have changed, the other references are either referred to the earlier pandemics or original works regarding the validity of the instruments that we have used in our study. Thus, we would like to keep these references although they are old but as said they refer to the original works.

Reviewer 4 Report

Thank you for the opportunity to review this study entitled “Coping, social support and loneliness during the COVID-19 pandemic and their effect on depression and anxiety: patients’ experiences in community mental health centers in Norway” (healthcare-1686690).

The study focused on the psychological effect of the COVID-19 pandemic, by exploring the associations between coping strategies, social support, loneliness, and worsening of mental health among patients in community mental health services. The research involved a sample of 164 patients.

In my opinion, the research topic is relevant, and the study is interesting. Parallelly, there are some issues that need to be addressed before the paper will be suitable for publication.

  • Abstract: the information about the sample should be deepened (Mean age and SD? Percentage of men and women?) to provide a clear picture of what will be presented in the paper.
  • Abstract: Please clarify already in the abstract that this is an online survey.
  • Introduction: In my opinion, it would be good to refer to trend or longitudinal studies, if any. Since the authors frame this study considering the impact that COVID-19 has on a psychological level on people, I suggest some research to propose a comprehensive framework in the introduction, which should be supplemented with further literature search by the authors:

- Hyland et al., 2021; doi: 10.1016/j.psychres.2021.113905.

- Gori & Topino, 2021; doi: 10.3390/ijerph18115651

- Wang et al., 2020; doi: 10.1016/j.bbi.2020.04.028

To find the suggested articles, the authors can use this source: https://www.doi.org/

  • Materials and methods: what was the response rate of those who expressed interest in participating in the study?
  • Statistical analysis section: which statistical software was used to perform the analyzes?
  • In the "discussion" section, the arguments relating to the absence of significant relationships with positive social support or adaptive coping strategies should be expanded. In fact, although these elements are highlighted in the literature as protective factors for mental health (e.g., DOI: 10.1192/bjp.bp.115.169094), the restrictions of the pandemic situation must also be taken into account, which for example make it difficult to obtain the desired social support (e.g., DOI: 10.1111/1468-5973.12380).

In general, I really enjoyed this paper, which seems to be well structured, interesting, and pleasant to read. In my opinion, after the authors make small changes, it will be ready to be published.

Author Response

Thank you for the opportunity to review this study entitled “Coping, social support and loneliness during the COVID-19 pandemic and their effect on depression and anxiety: patients’ experiences in community mental health centers in Norway” (healthcare-1686690).

The study focused on the psychological effect of the COVID-19 pandemic, by exploring the associations between coping strategies, social support, loneliness, and worsening of mental health among patients in community mental health services. The research involved a sample of 164 patients.

In my opinion, the research topic is relevant, and the study is interesting. Parallelly, there are some issues that need to be addressed before the paper will be suitable for publication.

  • Abstract: the information about the sample should be deepened (Mean age and SD? Percentage of men and women?) to provide a clear picture of what will be presented in the paper.

-We have inserted a sentence in the abstract as follows. Since the participants were asked to choose already defined 3 age categories, we cannot compute mean and SD for the age.

“Almost 51% were aged 31-50 years and 77% were females.”

  • Abstract: Please clarify already in the abstract that this is an online survey.

-We have made it clear in our revised abstract,

“Participants responded to an online questionnaire on corona-related questions…”

  • Introduction: In my opinion, it would be good to refer to trend or longitudinal studies, if any. Since the authors frame this study considering the impact that COVID-19 has on a psychological level on people, I suggest some research to propose a comprehensive framework in the introduction, which should be supplemented with further literature search by the authors:

- Hyland et al., 2021; doi: 10.1016/j.psychres.2021.113905.

- Gori & Topino, 2021; doi: 10.3390/ijerph18115651

- Wang et al., 2020; doi: 10.1016/j.bbi.2020.04.028

To find the suggested articles, the authors can use this source: https://www.doi.org/

-Thank you very much for suggesting us to add these articles from prospective studies. We have now referred to the first two articles (Hyland et al, Gori & Topino number 5-6).

  • Materials and methods: what was the response rate of those who expressed interest in participating in the study?

-The response rate was 31% as we have stated in the result section. “Of the 528 patients who received invitations and links to participate in the study, 164 responded (31%).”

  • Statistical analysis section: which statistical software was used to perform the analyzes?

-We have added the following  sentence under statistics.

We used SPSS version 27 for our data analyses.”

  • In the "discussion" section, the arguments relating to the absence of significant relationships with positive social support or adaptive coping strategies should be expanded. In fact, although these elements are highlighted in the literature as protective factors for mental health (e.g., DOI: 10.1192/bjp.bp.115.169094), the restrictions of the pandemic situation must also be taken into account, which for example make it difficult to obtain the desired social support (e.g., DOI: 10.1111/1468-5973.12380).

-Thanks for your suggestion, We have expanded and referred to the suggested articles.

In general, I really enjoyed this paper, which seems to be well structured, interesting, and pleasant to read. In my opinion, after the authors make small changes, it will be ready to be published.

Thank you very much for your encouraging remarks.

Round 2

Reviewer 2 Report

The study was always good, but this further information moves it from the interesting, to the academic data set of relavant. Thanks and well done.